# When Personalization Harms: Reconsidering the Use of Group Attributes for Prediction

## Abstract

Machine learning models often use group attributes to assign personalized pre-
dictions. In this work, we show that models that use group attributes can assign
unnecessarily inaccurate predictions to specific groups – i.e., that training a model
with group attributes can reduce performance for specific groups. We propose
formal conditions to ensure the "fair use" of group attributes in prediction models –
i.e., collective preference guarantees that can be checked by training one additional
model. We characterize how machine learning models can exhibit fair use due
to standard practices in specification, training, and deployment. We study the
prevalence of fair use violations in clinical prediction models. Our results highlight
the inability to resolve fair use violations, underscore the need to measure the
gains of personalization for all groups who provide personal data, and illustrate
actionable interventions to mitigate harm.

## 1  Introduction

Machine learning models are often used to support or automate decisions that affect people. In
medicine, for example, models diagnose illnesses [64, 31, 73], estimate survival rates [78], and
predict treatment response [41]. In such applications, medical decisions follow the ethical principles
of beneficence ("do the best") and non-maleficence ("do no harm") [8]. In turn, models that support
medical decisions are designed to perform as well as possible without inflicting harm. These principles
explain why so many clinical prediction models use *group attributes* that encode characteristics like
sex and age – i.e. characteristics that would be prohibited for models in lending or hiring. To predict
as well as possible on a heterogeneous population, models must encode all characteristics that could
tell people apart [47].

The prevalence of group attributes in prediction models reflects a need for *personalization*,[1] but
do personalized models that use group attributes improve performance for every group? In this
paper, we refer to this principle as *fair use*. Fair use enshrines the basic promise of personalization in
applications like precision medicine – i.e., that each person who reports personal characteristics should
expect a tailored performance gain in return. In prediction tasks with group attributes, this means
that every group should expect better performance from a *personalized model* that solicits group
membership compared to a *generic model* that does not. These gains should be *tailored*, meaning that
every group should prefer their personalized predictions over the personalized predictions assigned to
another group.

---

[1]Personalization is a term that encompasses a breadth of techniques that use personal data. Here, we
use it to describe approaches that target *groups* rather than *individuals* – i.e., "categorization" rather than
"individualization" as per the taxonomy of Fan & Poole [27].

Submitted to 2022 Trustworthy and Socially Responsible Machine Learning (TSRML 2022). Do not distribute.

| Group | Size | Error Rate | | Gain |
|---|---|---|---|---|
| $g$ | $n_g$ | $R(h_0)$ | $R_g(h_g)$ | $\Delta_g(h_g, h_0)$ |
| female, <30 | 48 | 38.1% | 26.8% | 11.3% |
| male, <30 | 49 | 23.9% | 26.7% | -2.8% |
| female, 30 to 60 | 307 | 30.3% | 29.1% | 1.2% |
| male, 30 to 60 | 307 | 15.4% | 15.2% | 0.2% |
| female, 60+ | 123 | 19.3% | 21.9% | -2.6% |
| male, 60+ | 181 | 11.0% | 8.2% | 2.8% |
| **Total** | 1152 | 20.4% | 19.4% | 1.0% |

**Figure 1:** Personalization can reduce performance for specific groups. We show the gains of personalization for a classifier to screen for obstructive sleep apnea (i.e., the `apnea` dataset in §4). We fit a personalized model $h_g$ and generic model $h_0$ with logistic regression, personalizing $h_g$ with a one-hot encoding of `sex` and `age_group`. As shown, personalization reduces training error from 20.4% to 19.4% but *increases* training error at for 2 groups: (`female, 60+`) and (`male, <30`). These effects are also present on test data.

Machine learning models are trained to use group attributes in ways that improve performance at a population level. In practice, this means that models trained with group attributes assign predictions that are unnecessarily inaccurate to specific groups due to routine decisions in model specification or model selection (see Figure 1). In many real-world applications, this drop in performance reflects harm. In clinical applications, for example, inaccurate predictions undermine medical decisions and health outcomes. This harm is silent and avoidable. Silent because fair use violations would only draw attention if model developers were to evaluate the *gains* of personalization for *intersectional* groups. Avoidable because a fair use violation shows that a group could receive better predictions from a generic model or a personalized model for another group; thus we can always resolve a fair use violation by assigning predictions from this better performing model.

Although many prediction models that use group attributes to assign personalized predictions, there is little awareness that this practice could reduce performance at a group level [see e.g., 2, 63]. Simply put, it is hard to imagine how a model that accounts for group membership can perform worse than a model that does not. Our goal in this paper is to expose this effect and lay the foundations to address it. To this end, we characterize how fair use violations arise, demonstrate their prevalence in real-world applications, and propose interventions to mitigate their harm. Specifically, the main contributions of our work include:

1. We propose formal conditions to ensure the fair use of group attributes in prediction models.

2. We characterize how common approaches to personalization in machine learning can produce personalized models to exhibit fair use violations. These "failure modes" delineate the root causes of fair use violations, and inform interventions that mitigate harm.

3. We conduct a comprehensive study on the gains of personalization in clinical prediction models for decision-making, ranking, and risk assessment. Our results demonstrate the prevalence of fair use violations across model classes and personalization techniques, and highlight the challenges of resolving these violations through changes to model development.

4. We present a case study on personalization for a model trained to predict mortality for patients with acute kidney injury. Our study shows how a fair use audit can safeguard against "race correction" in clinical prediction models, and facilitate targeted interventions that reduce harm (Appendix F).

## 2 Fair Use Guarantees

In this section, we present formal conditions for the fair use of group attributes in prediction. We provide notation and preliminaries for this section in Appendix A.

### 2.1 Fair Use

We start with Definition 1, which characterizes the fair use of a group attribute in terms of collective preference guarantees.

**Definition 1** (Fair Use). *A personalized model $h : \mathcal{X} \times \mathcal{G} \to \mathcal{Y}$ guarantees the fair use of a group attribute $\mathcal{G}$ if*

$$\Delta_{\boldsymbol{g}}(h_{\boldsymbol{g}}, h_0) \geq 0 \qquad \qquad \text{for all groups } \boldsymbol{g} \in \mathcal{G}, \qquad (1)$$

$$\Delta_{\boldsymbol{g}}(h_{\boldsymbol{g}}, h_{\boldsymbol{g}'}) \geq 0 \qquad \qquad \text{for all groups } \boldsymbol{g}, \boldsymbol{g}' \in \mathcal{G} \qquad (2)$$

Condition (1) captures *rationality* for group $\boldsymbol{g}$: a majority of group $\boldsymbol{g}$ prefers a personalized model $h_{\boldsymbol{g}}$ to a generic model $h_0$. Condition (2) captures *envy-freeness* for group $\boldsymbol{g}$: a majority of group $\boldsymbol{g}$ prefers their predictions to predictions personalized for any other group. These conditions enshrine minimal expectations of groups from a personalized model. Without rationality, a majority in some group would prefer the generic model. Without envy-freeness, a majority in some group would prefer the personalized predictions assigned to another group.

The fair use conditions in Definition 1 are collective, in that performance is measured over individuals in a group; and weak, in that the expected performance gain is non-negative – i.e., no group will be harmed. The conditions can be adapted to different prediction tasks by choosing a suitable risk metric. Since fair use conditions represent guarantees on the expected gains of personalization, a suitable metric should measure model performance exactly (c.f. a surrogate metric that we optimize to fit a model (see Figure 5 in Section 3). In classification tasks where we want accurate decisions, this would be the error rate. In tasks where we want reliable risk estimates, it would be the expected calibration error [54].

Personalized models that obey fair use guarantees incentivize groups to truthfully report group membership in deployment [see e.g., 39, 62, 30]. ]

## 2.2 Use Cases

Relevant use cases for fair use guarantees include:

*Protected Classes*: Models sometimes include group attributes that encode immutable characteristics due to application-specific norms or special provisions [see 44, 45]. For example, sex is a protected characteristic in employment law, but not in medicine [see e.g., 56, for a discussion on the use of sex to predict cardiovascular disease]. Likewise, U.S. regulations allow credit scores to use age if it does not harm older applicants [15]. In such cases, models should use these attributes in a way that leads to tailored performance gains for every group.

*Sensitive Data*: Models that use attributes like hiv status should guarantee a tailored improvement performance for the sensitive group, hiv = +. Otherwise, it would be better not to solicit this information in the first place as the information could inflicts harm when leaked [see e.g., 6].

*Self-Reported Data*: Certain kinds of models require users to report their data at prediction time [see e.g., self-report diagnostics 42, 67]. These models should obey fair use conditions to incentivize users to report their data truthfully (see Remark 2)

*Costly Data*: Group attributes can encode data collected at prediction time — e.g., an attribute like tumor_subtype whose value can only be determined by an invasive medical test. Models that ensure fair use with respect to tumor_subtype guarantee that patients with a specific type of tumor will not receive a less accurate prediction after undergoing the procedure.

## 3 Failure Modes of Personalization

In this section, we describe how common approaches to personalization can reduce performance for specific groups. Our goal is to highlight failure modes that apply to a broad range of prediction tasks. We pair each failure mode with toy examples, focusing on simple classification tasks that can be checked manually.[2]

## 3.1 Model Specification

We start with misspecification – i.e., when we fit models that cannot represent the role of group membership in the data distribution. A common form of misspecification occurs when we personalize

---

[2]In most cases, we train a linear classifier that minimizes the error rate on a *perfectly sampled* training dataset – i.e., where $\frac{1}{n} \sum_{i=1}^{n} 1[\boldsymbol{x}_i = \boldsymbol{x}, y_i = y, \boldsymbol{g}_i = \boldsymbol{g}] = \mathbb{P}(\boldsymbol{x}, y, \boldsymbol{g})$ for all $(\boldsymbol{x}, y, \boldsymbol{g}) \in \mathcal{X} \times \mathcal{Y} \times \mathcal{G}$. This condition ensures that the training error matches the test error.

| Group | Data | | Predictions | | Mistakes | | Gain |
|---|---|---|---|---|---|---|---|
| $g$ | $n_g^+$ | $n_g^-$ | $h_0$ | $h_g$ | $R_g(h_0)$ | $R_g(h_g)$ | $\Delta R_g(h_g, h_0)$ |
| young, female | 0 | 24 | − | + | 0 | 24 | −24 |
| young, male | 25 | 0 | − | + | 25 | 0 | 25 |
| old, female | 25 | 0 | − | + | 25 | 0 | 25 |
| old, male | 0 | 27 | − | − | 0 | 0 | 0 |
| **Total** | 50 | 51 | | | 50 | 24 | 26 |

**Figure 2:** Fair use violations due to model misspecification. Here, we are given $n^+ = 50$ positive examples and $n^- = 51$ negative examples for 2D classification task where $g \in \{\texttt{male}, \texttt{female}\} \times \{\texttt{old}, \texttt{young}\}$. We fit two linear classifiers: $h_0$, a generic model without group attributes, and $h_g$ a personalized model with a one-hot encoding. As shown, personalization reduces overall error from 50 to 24. However, not all groups benefit from personalization: (young, female) now receives less accurate predictions while (old, male) receives no gain. Here, $h_g$ also violates envy-freeness for (young, female) as individuals in this group would receive more accurate predictions by misreporting their group membership as (old, male).

110 simple models using a one-hot encoding. In such cases, models exhibit fair use violations on data
111 distributions that exhibit intersectionality (see Figure 2). Consider, for example, a logistic regression
112 model with a one-hot encoding that assigns higher risk to patients who are `old` and to patients who
113 are `male`. This would lead to a fair use violation for patients who are `old` *and* `male` if their true risk
114 were lower than either group alone.

115 Misspecification can also arise due to a failure to account for group-specific interaction effects – e.g.,
116 instances where group attributes act as mediator or moderator variables [see e.g., 7]. In Figure 3, we
117 show an example that exhibits the hallmarks of personalization: a generic model performs poorly
118 on "heterogeneous" groups $A$ and $C$, and a personalized model that accounts for group membership
119 improves overall performance by assigning more accurate predictions to $A$ and $C$. In this case, the
120 resulting model exhibits a fair use violation for group $B$ because a generic model performs as well as
121 possible for group $B$.

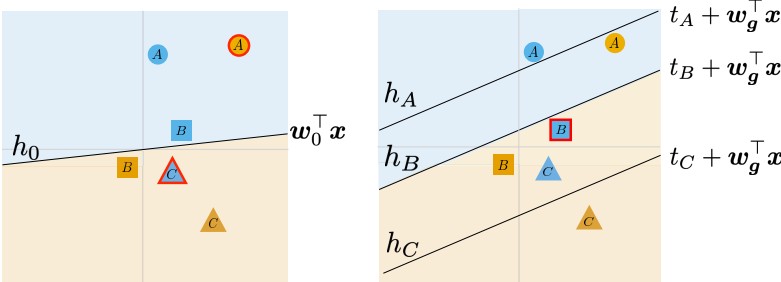

**Figure 3:** Fair use violation resulting from model misspecification. We consider a 2D classification task with heterogeneous groups $g = \{A, B, C\}$ where an ideal model should assign a personalized intercept to each group *and* a personalized slope to group $B$. In this case, a personalized model with a one-hot encoding would fit a personalized intercept for each group, but fail to fit a personalized slope for group $B$. The personalized model would improve overall performance by assigning more accurate predictions to groups $A$ and $C$. However, it would result in a fair use violation by performing *worse* for group $B$.

122 In practice, we can avoid these issues by either fitting models that are rich enough to capture these
123 effects, or by training a separate model for each group. Both are challenging in tasks with multiple
124 groups as we must either specify interactions for each group, or fit models using a limited amount of
125 data for each group.

## 3.2 Model Selection

127 Model development often involves choosing one model from a family of candidate models – e.g.,
128 when we set a regularization penalty to avoid overfitting, or choose a subset of variables to improve
129 usability. Common criteria for model selection consist choosing a model on the basis of population-
130 level performance [e.g., mean K-CV test error 4]. In practice, this choice can lead to models that
131 reduce performance for a specific group. We demonstrate this effect in Figure 4. The example

highlights how fair use violations may be unavoidable in settings where we are forced to assign predictions with a single model – as there may not exist a model that ensure fair use for all groups.

### 3.3 Other Failure Modes & Discussion

Work in personalization naturally presumes that fitting a model with group attributes will provide a uniform performance gain to all groups. In practice, however, this only holds under restrictive assumptions. We include a similar discussion of other failure modes along with examples in Appendix E including: training with a surrogate loss function; generalization; and dataset shifts. The failure models that we have covered in this section are chosen since they motivate potential interventions for model development. For example, one could avoid the fair use violations in Figure 2 by using an intersectional one-hot encoding, and avoid violations across across all cases by training decoupled models.

## 4 Empirical Study

In this section, we study fair use in clinical prediction models – i.e. models that routinely include group attributes where fair use violations inflict harm. Our goals are to measure the prevalence of fair use violations and to evaluate how these change as a result of interventions in model development. We attach all software to reproduce the results in this section to our submission, and include additional details on our setup and additional experimental results in the supplement.

### 4.1 Setup

We work with 6 datasets for clinical prediction tasks (see Table 1). We split each dataset into a training sample (80%) to fit models, and a test sample (20%) to evaluate the gains of personalization. We use the training data from each dataset to fit 9 kinds of personalized models. Each personalized model belongs to one of 3 model classes: *logistic regression* (LR), *random forests* (RF), and *neural nets* (NN); and accounts for group membership using one of 3 personalization techniques.

The three personalization techniques being: *One-hot Encoding* (1Hot): We fit a model with dummy variables for each group attribute, *Intersectional Encoding* (All): We fit a model with dummy variables for each intersectional group, and *Decoupling* (DCP): We fit a model for each intersectional group using its own data. The three techniques represent increasingly complex ways to account for group membership where complexity is measured by the interactions between group attributes and other features: 1Hot reflect no interactions; All reflect interactions between group attributes; and DCP reflects all possible attributes between group attributes and features.

We evaluate the gains of personalization for each model in terms of three performance metrics: (1) *error rate*, which reflects the accuracy of yes-or-no predictions [for a diagnostic test, e.g., 26]; (2) *expected calibration error* (ECE), which measures the reliability of risk predictions [for a medical risk score, e.g., 13]; (3) *area under ROC curve* (AUC), which measures accuracy in ranking [for a prioritization tool, e.g., 77].

### 4.2 Results

We summarize our results for logistic regression in Table 1 and for other model classes in Appendix G.

**On Prevalence**   Our results show that personalized models can improve performance at a population level yet reduce performance for specific groups. These fair use violations arise across datasets, personalization techniques, and model classes. Consider the standard configuration used to develop clinical prediction models – i.e., a logistic regression model with a one-hot encoding of group attributes (LR+1Hot). Here, we find that at least one group experiences a statistically significant fair use violation in terms of error on 4/6 datasets (5/6 for AUC and ECE).

**On Personalization Techniques**   Our results show that there is no one personalization technique that minimizes fair use violations. In Table 1, for example, the best personalization technique for `cardio_eicu` is intersectional encoding while the best personalization technique for `mortality`

| Dataset | Metrics | Test AUC | | | Test ECE | | | Test Error | | |
|---|---|---|---|---|---|---|---|---|---|---|
| | | 1Hot | All | DCP | 1Hot | All | DCP | 1Hot | All | DCP |
| apnea | Personalized | 0.750 | 0.750 | 0.803 | 7.5% | 5.5% | 7.2% | 34.2% | 33.8% | 26.2% |
| $n = 1152, d = 26$ | Gain | 0.001 | 0.000 | 0.053 | -1.5% | 0.6% | -1.1% | -1.0% | -0.7% | 7.0% |
| $\mathcal{G} = \{\texttt{age},\texttt{sex}\}$ | Best/Worst Gain | 0.002 / -0.001 | 0.001 / -0.016 | 0.119 / -0.005 | 0.7% / -7.1% | 0.7% / -4.6% | 1.7% / -6.6% | 0.0% / -9.9% | 1.8% / -7.8% | 21.7% / -7.8% |
| $m = 6$ | Rat. Gains/Viols | 1/2 | 1/4 | 4/0 | 1/3 | 1/3 | 2/2 | 0/6 | 0/5 | 4/1 |
| [66] | EF Gains/Viols | 0/0 | 0/0 | 3/0 | 0/3 | 0/3 | 4/0 | 0/6 | 0/5 | 4/1 |
| cardio_eicu | Personalized | 0.768 | 0.767 | 0.762 | 4.4% | 4.6% | 8.9% | 29.1% | 29.1% | 29.5% |
| $n = 1341, d = 49$ | Gain | 0.000 | -0.001 | -0.007 | 0.4% | 0.2% | -4.1% | -0.4% | -0.4% | -0.9% |
| $\mathcal{G} = \{\texttt{age},\texttt{sex}\}$ | Best/Worst Gain | 0.002 / -0.001 | 0.001 / -0.001 | 0.094 / -0.099 | 1.6% / -1.5% | 0.9% / -0.2% | -1.1% / -6.3% | 0.0% / -3.1% | 0.2% / -3.1% | 12.9% / -8.9% |
| $m = 4$ | Rat. Gains/Viols | 2/2 | 2/1 | 1/2 | 2/1 | 1/0 | 0/4 | 0/2 | 1/2 | 2/2 |
| [60] | EF Gains/Viols | 0/0 | 0/0 | 3/1 | 0/2 | 0/2 | 1/1 | 0/3 | 0/3 | 3/1 |
| cardio_mimic | Personalized | 0.854 | 0.854 | 0.870 | 2.1% | 2.3% | 2.3% | 23.3% | 23.4% | 21.4% |
| $n = 5289, d = 49$ | Gain | 0.001 | 0.001 | 0.017 | -0.4% | -0.5% | -0.6% | 0.3% | 0.3% | 2.2% |
| $\mathcal{G} = \{\texttt{age},\texttt{sex}\}$ | Best/Worst Gain | 0.001 / -0.001 | 0.001 / -0.000 | 0.051 / 0.006 | 0.5% / 0.4% | 0.6% / -0.2% | 0.6% / -2.3% | 0.9% / -0.1% | 0.9% / -0.1% | 7.6% / -0.2% |
| $m = 4$ | Rat. Gains/Viols | 2/1 | 2/1 | 4/0 | 4/0 | 3/0 | 1/2 | 3/0 | 3/0 | 3/0 |
| [38] | EF Gains/Viols | 0/0 | 0/0 | 4/0 | 1/3 | 0/1 | 3/1 | 0/3 | 0/3 | 4/0 |
| heart | Personalized | 0.870 | 0.846 | 0.817 | 8.4% | 17.8% | 17.5% | 19.7% | 19.7% | 15.8% |
| $n = 181, d = 26$ | Gain | -0.007 | -0.030 | -0.060 | 2.8% | -6.6% | -6.3% | -1.3% | -1.3% | 2.6% |
| $\mathcal{G} = \{\texttt{sex},\texttt{age}\}$ | Best/Worst Gain | 0.007 / -0.031 | 0.024 / -0.050 | 0.039 / -0.190 | 4.4% / -0.6% | -1.8% / -3.1% | 10.1% / -4.6% | 0.0% / -6.1% | 0.0% / -12.1% | 10.6% / -8.4% |
| $m = 4$ | Rat. Gains/Viols | 1/1 | 1/1 | 0/3 | 2/1 | 0/4 | 2/1 | 0/1 | 0/3 | 3/1 |
| [17] | EF Gains/Viols | 0/0 | 0/0 | 1/2 | 0/2 | 0/3 | 2/2 | 0/1 | 0/1 | 2/1 |
| mortality | Personalized | 0.848 | 0.848 | 0.880 | 2.0% | 2.1% | 2.5% | 23.6% | 23.4% | 20.2% |
| $n = 25366, d = 468$ | Gain | 0.000 | 0.001 | 0.033 | 0.2% | 0.1% | -0.3% | -0.2% | -0.0% | 3.2% |
| $\mathcal{G} = \{\texttt{age},\texttt{sex}\}$ | Best/Worst Gain | 0.005 / -0.001 | 0.005 / -0.000 | 0.111 / -0.012 | 1.5% / 0.1% | 2.6% / -0.3% | 11.2% / -2.4% | 0.8% / -2.5% | 2.1% / -0.4% | 20.1% / -0.5% |
| $m = 6$ | Rat. Gains/Viols | 3/3 | 3/2 | 6/0 | 5/0 | 5/1 | 3/2 | 2/4 | 3/2 | 5/1 |
| [38] | EF Gains/Viols | 0/0 | 0/0 | 6/0 | 1/1 | 3/2 | 5/1 | 0/4 | 1/4 | 6/0 |
| saps | Personalized | 0.890 | 0.890 | 0.888 | 1.5% | 1.5% | 2.0% | 18.9% | 18.9% | 18.5% |
| $n = 7797, d = 36$ | Gain | 0.001 | 0.001 | -0.001 | 0.1% | 0.1% | -0.4% | 0.0% | 0.0% | 0.4% |
| $\mathcal{G} = \{\texttt{hiv},\texttt{age}\}$ | Best/Worst Gain | 0.014 / -0.000 | 0.014 / -0.001 | 0.017 / -0.246 | 2.8% / -1.5% | 2.4% / -0.6% | 9.4% / -19.1% | 19.0% / -10.4% | 0.8% / -10.4% | 3.5% / -23.3% |
| $m = 4$ | Rat. Gains/Viols | 1/1 | 1/1 | 2/2 | 2/0 | 2/1 | 2/2 | 2/1 | 1/3 | 2/1 |
| [3] | EF Gains/Viols | 0/0 | 0/0 | 2/1 | 2/2 | 2/2 | 3/1 | 1/1 | 2/2 | 2/2 |

**Table 1:** Performance of personalized logistic regression models on all datasets. We show the gains of personalization in terms of test AUC, ECE, and error. We report: model performance at the population level, the overall gain of personalization, the range of gains over $m$ intersectional groups, and the number of rationality and envy-freeness gains/violations (evaluated using a bootstrap hypothesis test at a 10% significance level).

was decoupling. These strategies change across model classes – as the corresponding strategies for neural networks for cardio_eicu and mortality are decoupling and using an intersectional encoding, respectively (see Appendix G). In general, even strategies that exhibit few violations can fail critically. For example, LR+DCP for saps leads to a 10% increase in error for HIV+ & >30. Overall, these results suggest that the most consistent way to avoid the harm from a fair use violation is to check.

**On Interventions in Model Development** Our results show that routine decisions in model development can produce considerable differences in group-level performance and fair use violations. This suggests that if we are able to spot fair use violations, we may be able to minimize them by "interventions" to model development. In light of this, we consider interventions that address the failure modes in Section 3 – e.g., using an intersectional one-hot encoding, training decoupled models, and equalizing sample sizes.

In general, we find that applying these strategies can minimize fair use violations often. For example, we can eliminate all fair use violations for cardio_mimic in our standard configuration by training decoupled models. However, there is no "best" intervention that consistently resolve these violations. Typically, this is because an intervention that resolves a violation for one group will precipitate a violation for others. In cardio_eicu, for instances, a logistic regression model fit with a onehot encoding will exhibit a violation on old males. Switching an intersectional encoding will fix this violation but introduce a new one for old females.

**On the Reliability of Gains & Violations** Our results underscore the need for reliable procedures to discover fair use violations or claim gains from personalization. We can often find detectable instances of benefit or harm. For example, we find that on saps in our default configuration that we detect a gain from personalization for patients who are HIV negative and older than 30. Additionally, in cardio_eicu when training LR+All we detect a fair use violation for patients who are old females (see e.g., Rat Gains/Violations in Table 1). One actionable finding from an evaluation of the gains of personalization is a group does not experience a meaningful gain nor harm due to personalization. In such cases, one may wish to intervene to avoid soliciting unnecessary data: when group attributes encode information that is sensitive or that must be collected at prediction time (e.g., hiv_status or tumor_subtype), we may prefer to avoid soliciting information that is demonstrably useful for prediction.

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

## A  Notation

Below we provide a table that consolidates and describes the notation used throughout the paper.

| Symbol | Meaning |
| --- | --- |
| $\boldsymbol{x}_i = (x_{i,1}, x_{i,2}, \ldots, x_{i,d})$ | feature vector of example $i$ |
| $y_i \in \mathcal{Y}$ | label of example $i$ |
| $\boldsymbol{g}_i \in \{g_{i,1}, g_{i,2}, \ldots, g_{i,k}\}$ | group membership of example $i$ |
| $\mathcal{G} = \mathcal{G}_1 \times \mathcal{G}_2 \times \ldots \times \mathcal{G}_k$ | space of group attributes |
| $m = |\mathcal{G}|$ | number of intersectional groups |
| $n_{\boldsymbol{g}} := \sum \mathbb{1}[\boldsymbol{g}_i = \boldsymbol{g}]$ | number of examples of group $\boldsymbol{g} \in \mathcal{G}$ |
| $n_{\boldsymbol{g}}^{+} := \sum \mathbb{1}[\boldsymbol{g}_i = \boldsymbol{g},\, y_i = +1]$ | number of examples of group $\boldsymbol{g} \in \mathcal{G}$ with $y_i = +1$ |
| $n_{\boldsymbol{g}}^{-} := \sum \mathbb{1}[\boldsymbol{g}_i = \boldsymbol{g},\, y_i = -1]$ | number of examples of group $\boldsymbol{g} \in \mathcal{G}$ with $y_i = -1$ |
| $\mathcal{H}_0$ | hypothesis class of generic model |
| $\mathcal{H}$ | hypothesis class of personalized models |
| $h_0 \in: \mathcal{X} \to \mathcal{Y}$ | generic model |
| $h : \mathcal{X} \times \mathcal{G} \to \mathcal{Y}$ | personalized model |
| $h_{\boldsymbol{g}} : \mathcal{X} \times \mathcal{G} \to \mathcal{Y}$ | personalized classifier where group membership is reported truthfully (as $\boldsymbol{g}$) |
| $R_{\boldsymbol{g}}(h_{\boldsymbol{g}'})$ | true risk of model $h$ of group $\boldsymbol{g}$ if they report $\boldsymbol{g}'$ |
| $\hat{R}_{\boldsymbol{g}}(h_{\boldsymbol{g}'})$ | empirical risk of model $h$ of group $\boldsymbol{g}$ if they report $\boldsymbol{g}'$ |
| $\Delta_{\boldsymbol{g}}(h, h')$ | gain (i.e., reduction in true risk) for group $\boldsymbol{g}$ when using $h$ rather than $h'$ |
| $\Delta_{\boldsymbol{g}}(h_{\boldsymbol{g}}, h_0)$ | rationality gap for group $\boldsymbol{g}$ (performance gain when reporting $\boldsymbol{g}$ as opposed to concealing it) |
| $\Delta_{\boldsymbol{g}}(h_{\boldsymbol{g}}, h_{\boldsymbol{g}'})$ | rationality gap for group $\boldsymbol{g}$ (performance gain when reporting $\boldsymbol{g}$ as opposed to concealing it) |

**Table 2:** Notation

### Preliminaries

We start with a dataset of $n$ examples $(\boldsymbol{x}_i, y_i, \boldsymbol{g}_i)_{i=1}^n$, where each example consists of a feature vector $\boldsymbol{x}_i = [x_{i,1}, \ldots, x_{i,d}] \in \mathbb{R}^d$, a label $y_i \in \mathcal{Y}$, and a vector of $k$ categorical *group attributes* $\boldsymbol{g}_i = [\boldsymbol{g}_{i,1}, \ldots, \boldsymbol{g}_{i,k}] \in \mathcal{G}_1 \times \ldots \times \mathcal{G}_k = \mathcal{G}$ – e.g., $\boldsymbol{g}_i = [\texttt{female}, \texttt{age} \geq \texttt{60}, \texttt{blood\_type = O+}]$. We refer to $\boldsymbol{g}_i$ as the *group membership* of $i$ and to the set $\{i \mid \boldsymbol{g}_i = \boldsymbol{g}\}$ as *group $\boldsymbol{g}$*. We let $n_{\boldsymbol{g}} := |\{i \mid \boldsymbol{g}_i = \boldsymbol{g}\}|$ denote the number of examples in group $\boldsymbol{g}$, and let $m := |\mathcal{G}|$ denote the number of intersectional groups.

We use the data to fit a *personalized* model that uses group attributes $h : \mathcal{X} \times \mathcal{G} \to \mathcal{Y}$; and a *generic* model that does not $h_0 : \mathcal{X} \to \mathcal{Y}$. We fit both models via empirical risk minimization with a loss function $\ell : \mathcal{Y} \times \mathcal{Y} \to \mathbb{R}_+$, using $\hat{R}(h)$ and $R(h)$ to denote the empirical risk and true risk, respectively. We assume that the personalized and generic models represent the best models trained on datasets with group attributes $(\boldsymbol{x}_i, y_i, \boldsymbol{g}_i)_{i=1}^n$ and without them $(\boldsymbol{x}_i, y_i)_{i=1}^n$:

$$h \in \underset{h \in \mathcal{H}}{\operatorname{argmin}}\, \hat{R}(h) \qquad h_0 \in \underset{h \in \mathcal{H}_0}{\operatorname{argmin}}\, \hat{R}(h)$$

We evaluate the gains of personalization for a personalized model $h$ for each group. As part of this evaluation, we examine how the performance of $h$ for group $\boldsymbol{g}$ changes when they are assigned predictions that are personalized for another group $\boldsymbol{g}'$ – i.e., the predictions that group $\boldsymbol{g}$ would receive by "misreporting" their group membership as $\boldsymbol{g}'$. We represent this formally by using $h_{\boldsymbol{g}'} := h(\cdot, \boldsymbol{g}')$ to denote a personalized model where group attributes are fixed to $\boldsymbol{g}'$. Given a personalized model $h$, we measure its *empirical risk* and *true risk* for group $\boldsymbol{g}$ when they report group membership as $\boldsymbol{g}'$ as:

$$\hat{R}_{\boldsymbol{g}}(h_{\boldsymbol{g}'}) := \frac{1}{n_{\boldsymbol{g}}} \sum_{i:\boldsymbol{g}_i = \boldsymbol{g}} \ell\left(h(\boldsymbol{x}_i, \boldsymbol{g}'), y_i\right) \qquad R_{\boldsymbol{g}}(h_{\boldsymbol{g}'}) := \mathbb{E}\left[\ell\left(h(\boldsymbol{x}, \boldsymbol{g}'), y\right) \mid \mathcal{G} = \boldsymbol{g}\right].$$

We assume that groups prefer models that assign more accurate predictions as measured in terms of true risk. We express the preferences of group $\boldsymbol{g}$ between $h$ and $h'$ using the *gain* measure $\Delta_{\boldsymbol{g}}(h, h') := R_{\boldsymbol{g}}(h) - R_{\boldsymbol{g}}(h')$.

## B  Related Work

Our work is related to several streams of research in algorithmic fairness. We propose to check the quality of personalization using preference-based notions of fairness [75, 68, 43, 70, 20]. We focus

on intersectional groups [c.f., 40, 35], which are more granular than those considered in the literature yet large enough to estimate performance [c.f., 22, 5]. We study models that use group attributes to assign more accurate predictions over a heterogeneous population. Several works highlight the need to account for group membership [75, 23, 16, 46, 50, 72], observing that it is otherwise impossible for a model to achieve *parity* – i.e., to perform equally well for all groups [33, 74, 76, 28, 1, 55, 14]. Parity-based methods are ill-suited for personalization since they equalize performance by reducing performance for groups for who the model performs well, rather than improving performance for groups for who the model performs poorly [50, 36, 58, 51, 52].

We study personalization in models that encode personal characteristics through categorical attributes, which are widely used across medicine, consumer finance, and criminal justice (see use cases in §2). In medicine, for example, many models are fit using logistic regression with a one-hot encoding of categorical attributes [63, 71, 25]. Existing work that evaluates the gain of personalization often does so at population-level rather at the level of group who provide personal data [37, 65]. This population-level focus characterizes technical work in this area: recent methods use categorical attributes to improve population-level performance by accounting for heterogeneity – e.g., by automatically including higher-order interaction effects [11, 49, 69] or recursively partitioning data [24, 12, 10, 9].

## C   Truthful Self-Reporting

**Remark 2** (Truthful Self-Reporting). *Consider a prediction task where each person reports their group membership to a personalized model. Let $\boldsymbol{r}_i$ denote the self-reported group membership of person $i$ where:*

$$\boldsymbol{r}_i = \boldsymbol{g}_i \Leftrightarrow i \text{ reports truthfully} \qquad \boldsymbol{r}_i \in \mathcal{G} \setminus \{\boldsymbol{g}_i\} \Leftrightarrow i \text{ misreports} \qquad \boldsymbol{r}_i =? \Leftrightarrow i \text{ withhold}$$

*If a personalized model $h : \mathcal{X} \times \mathcal{G} \to \mathcal{Y}$ guarantees the fair use of a group attribute $\mathcal{G}$ then each person would opt to truthfully report as this strategy would maximize their expected performance:*

$$\boldsymbol{g}_i \in \underset{\boldsymbol{r}_i \in G \cup \{?\}}{\mathrm{argmin}} \ \mathbb{E}\left[\ell\left(h(\boldsymbol{x}, \boldsymbol{r}_i), y_i\right) \mid \mathcal{G} = \boldsymbol{g}_i\right].$$

Truthful reporting incentives reflect basic principles regarding *consent* in data privacy rights. In effect, a personalized model that exhibits a fair use violation for a specific group uses their group membership in a way that is coercive. If a group were allowed to report their personal information to the model at prediction time, they would opt to withhold or misreport this information. With respect to Definition 1, rationality ensures that a majority of $\boldsymbol{g}$ prefer to report group membership rather than withhold it. Envy-freeness ensures that a majority of group $\boldsymbol{g}$ prefer to report group membership rather than misreport it.

## D   Testing & Verification

Point estimates of the gains of personalization are not reliable, especially for small groups. In a prediction task where a personalized model performs 5% worse than a generic model, a 5% drop could represent 5 mistakes for a group with 100 samples, or 200 mistakes for a group with 4000 samples. Measuring the statistical significance of gains can help us distinguish between such cases and inform our use of group attributes. In some applications, a significant fair use violation could warrant the need for a new model. In others, we may wish to ensure a significant gain to use a group attribute in the first place.

In practice, we check for a rationality violation using a one-sided hypothesis test of the form:

$$H_0 : R(h_0) - R(h_{\boldsymbol{g}}) \leq 0 \qquad H_A : R(h_0) - R(h_{\boldsymbol{g}}) > 0$$

Here, the null hypothesis $H_0$ assumes that group $\boldsymbol{g}$ prefers $h_{\boldsymbol{g}}$ to $h_0$ by default. Thus, we reject $H_0$ when there is enough evidence to support a rationality violation for $\boldsymbol{g}$ in a held-out dataset.

We can use an inverted setup where $H_A : R(h_0) - R(h_{\boldsymbol{g}}) < 0$ to check for gains from personalization.The testing procedure varies based on the performance metric used to evaluate the gains of personalization. In general, we can apply a bootstrap hypothesis test [18]. In some cases, there exist more powerful tests for specific performance metrics [see e.g., the McNemar test for accuracy 19]. We can repeat these tests across multiple groups to check for envy-freeness, or to check for all conditions in Definition 1. In the latter regimes, we can control for the false discovery rate using a standard Bonferroni correction[21], which is suitable even for non-independent tests.

## E   Failure Modes of Personalization

In this Appendix, we describe additional mechanisms that lead personalized models to exhibit fair use violations. The mechanisms below reflect failure modes that arise in later stages of the machine learning pipeline, and that are more difficult to address through interventions.

### E.1   Model Selection

### E.2   ERM with a Surrogate Loss Function

Consider a setting where we want a personalized model that maximizes classification accuracy – i.e., one that minimizes the 0–1 loss. If we fit this classifier using a linear SVM – e.g., by solving an ERM problem that optimizes the hinge loss – the approximation error between the 0-1 loss and the hinge loss can produce a fair use violation (see Figure 5). This example is specifically designed to avoid fair use violations that stem from model misspecification.

| $(g, x_1, x_2)$ | Data $n^+$ | $n^-$ | Generic $h_0 = h_0(x_1) = h_0(x_2)$ Pred. $h_0$ | Mistakes $R(h_0)$ | Personalized Model (Selected) $h_S(x_1, g)$ Pred. $h_S$ | Error $R(h_S)$ | Gain $\Delta R_g(h_0, h_S)$ | Personalized Model (Discarded) $h_D(x_2, g)$ Pred. $h_D$ | Error $R(h_D)$ | Gain $\Delta R_g(h_0, h_D)$ |
|---|---|---|---|---|---|---|---|---|---|---|
| $(0,0,0)$ | 0 | 30 | $-$ | 0 | $-$ | 0 | 0 | $+$ | 30 | $-30$ |
| $(0,0,1)$ | 0 | 0 | $-$ | 0 | $-$ | 0 | 0 | $-$ | 0 | 0 |
| $(0,1,0)$ | 0 | 20 | $-$ | 0 | $+$ | 20 | $-20$ | $+$ | 20 | $-20$ |
| $(0,1,1)$ | 0 | 0 | $-$ | 0 | $+$ | 0 | 0 | $-$ | 0 | 0 |
| $(1,0,0)$ | 25 | 0 | $-$ | 25 | $+$ | 0 | 25 | $+$ | 0 | 25 |
| $(1,0,1)$ | 0 | 0 | $-$ | 0 | $+$ | 0 | 0 | $-$ | 0 | 0 |
| $(1,1,0)$ | 15 | 0 | $-$ | 15 | $+$ | 0 | 15 | $+$ | 0 | 15 |
| $(1,1,1)$ | 0 | 0 | $-$ | 0 | $-$ | 0 | 0 | $-$ | 0 | 0 |
| Total | 40 | 50 | | 40 | | 35 | 5 | | 50 | $-10$ |
| $g = 0$ | 0 | 50 | | 0 | | 20 | $-20$ | | 50 | $-50$ |
| $g = 1$ | 40 | 0 | | 40 | | 15 | 25 | | 0 | 40 |

**Figure 4:** Standard model selection criteria can lead to fair use violations. We consider a 2D classification task with two groups $g \in \{0, 1\}$ where we need a model that can use at most one of the binary attributes $(x_1, x_2) \in \{0, 1\}^2$. We fit a generic model and a personalized model with a one-hot encoding of group membership choosing the variable that minimizes the overall error rate. Here, each group performs better under different choices, defaulting to choice that benefits the majority group.

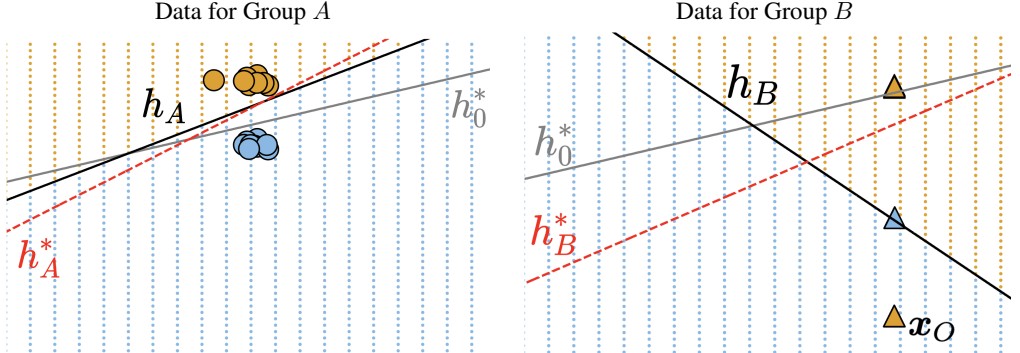

**Figure 5:** Fair use violations resulting from the use of surrogate loss function in ERM. Here, we are given data for classification task with features $\boldsymbol{x} = (x_1, x_2)$ and a group attribute $\boldsymbol{g} = \{A, B\}$. We fit a linear SVM $h_{\boldsymbol{g}}$ by optimizing the hinge loss for a prediction task where evaluate the gains of personalization in terms of the error rate (i.e., 0-1 loss). In this case, the personalized model produces a fair use violation for Group $B$ due to an outlier $\boldsymbol{x}_O$. We plot the data for group $A$ and group $B$ separately. Each plot shows the generic classifier ($h_0$; grey) and the personalized classifiers for the corresponding group ($h_A$ or $h_B$; black). As a baseline for comparison, we show the personalized models that we would obtain by optimizing an exact loss function (i.e., 0-1 loss, which matches the performance metric that we use to evaluate the gains for personalization). As shown, we would expect to avoid this violation had we fit a model by optimizing the 0–1 loss directly.

### E.3  Generalization & Dataset Shifts

Fair use violations can arise in deployment. Small samples may significantly distort the relative prevalence of each group, leading standard empirical risk minimization to fit a suboptimal generic model or personalized model (see Figure 6). Fair use violations can also arise as a result of changes in the data distribution [i.e., dataset shift  61, 29, 32] (see Figure 7)

| Group | | Training Data | | Data Distribution | | Model Predictions | | Observed Performance | | | True Performance | | |
|---|---|---|---|---|---|---|---|---|---|---|---|---|---|
| $g_1$ | $g_2$ | $n^+$ | $n^-$ | $n^+$ | $n^-$ | $h_0(\boldsymbol{x})$ | $h_{\boldsymbol{g}}(\boldsymbol{x},\boldsymbol{g})$ | $R_{\boldsymbol{g}}(h_0)$ | $R_{\boldsymbol{g}}(h_{\boldsymbol{g}})$ | $\Delta_{\boldsymbol{g}}(h_{\boldsymbol{g}},h_0)$ | $R_{\boldsymbol{g}}(h_0)$ | $R_{\boldsymbol{g}}(h_{\boldsymbol{g}})$ | $\Delta_{\boldsymbol{g}}(h_{\boldsymbol{g}},h_0)$ |
| 0 | 0 | 65 | 60 | 130 | 120 | + | + | 60 | 60 | 0 | 120 | 120 | 0 |
| 1 | 0 | 60 | 65 | 120 | 130 | + | − | 65 | 60 | 5 | 130 | 120 | 10 |
| 0 | 1 | 60 | 65 | 130 | 120 | + | − | 65 | 60 | 5 | 120 | 130 | −10 |
| 1 | 1 | 70 | 55 | 140 | 110 | + | + | 55 | 55 | 0 | 110 | 110 | 0 |
| | Total | 255 | 245 | 520 | 480 | − | N/A | 245 | 235 | 10 | 480 | 470 | 0 |

**Figure 6:** Fair use violations can arise when personalizing models on small samples. Here, we show a 2D classification task in which a personalized model only exhibits fair use violations in deployment. Here, group $(1,0)$ experiences an gain once the model is deployment. In contrast, group $(0,1)$ experiences a fair use violation as a result of sampling error.

| Group | | Training Data | | True Distribution | | Model Predictions | | Train Performance | | | True Performance | | |
|---|---|---|---|---|---|---|---|---|---|---|---|---|---|
| $g_1$ | $g_2$ | $n^+$ | $n^-$ | $n^+$ | $n^-$ | $h_0(\boldsymbol{x})$ | $h_{\boldsymbol{g}}(\boldsymbol{x},\boldsymbol{g})$ | $R_{\boldsymbol{g}}(h_0)$ | $R_{\boldsymbol{g}}(h_{\boldsymbol{g}})$ | $\Delta_{\boldsymbol{g}}(h_{\boldsymbol{g}},h_0)$ | $R_{\boldsymbol{g}}(h_0)$ | $R_{\boldsymbol{g}}(h_{\boldsymbol{g}})$ | $\Delta_{\boldsymbol{g}}(h_{\boldsymbol{g}},h_0)$ |
| 0 | 0 | 20 | 0 | 20 | 0 | + | + | 0 | 0 | 0 | 0 | 0 | 0 |
| 1 | 0 | 5 | 25 | 5 | 25 | + | − | 25 | 5 | 20 | 25 | 5 | 20 |
| 0 | 1 | 5 | 25 | 30 | 25 | + | − | 25 | 5 | 20 | 20 | 30 | −10 |
| 1 | 1 | 20 | 0 | 20 | 0 | + | + | 0 | 0 | 0 | 0 | 0 | 0 |
| | Total | 50 | 50 | 75 | 45 | + | N/A | 50 | 10 | 40 | 45 | 35 | 10 |

**Figure 7:** Label shift produces a fair use violation. Here, we train a linear classifier on a dataset with [one binary feature and one binary group attribute]. As shown, personalization leads to overall improvement reducing aggregate reduce from 50 to 24 and group-specific improvement on the training data. However, not all groups perform equally well in deployment. While groups $(0,1)$ and $(1,1)$ see improvements, a violation (red) occurs for group $(1,0)$ due to the label shift where positive examples are no longer present meanwhile they were the majority in the training data.

| Group | Test AUC | | Interventions | | Test Error | | Intervention | | Test ECE | | Interventions | |
|---|---|---|---|---|---|---|---|---|---|---|---|---|
| $g$ | $R_g(h_g)$ | $\Delta_g$ | Assign $h_0$ | Assign $h_{dcp}$ | $R_g(h_g)$ | $\Delta_g$ | Assign $h_0$ | Assign $h_g^{dcp}$ | $R_g(h_g)$ | $\Delta_g$ | Assign $h_0$ | Assign $h_g^{dcp}$ |
| female, black | 0.463 | 0.024 | 0.024 | 0.334 | 52.2% | 6.8% | 6.8% | 37.3% | 31.6% | 2.3% | 2.3% | 12.3% |
| female, white | 0.846 | 0.004 | 0.004 | 0.004 | 21.7% | 2.0% | 2.0% | 2.0% | 10.2% | 1.9% | 1.9% | 2.1% |
| female, other | 0.860 | -0.003 | 0.000 | 0.057 | 25.5% | 1.3% | 1.3% | 14.8% | 15.5% | 0.9% | 0.9% | 5.0% |
| male, black | 0.767 | -0.001 | 0.000 | 0.104 | 34.0% | -5.2% | 0.0% | 15.6% | 20.1% | -2.0% | 0.0% | 4.9% |
| male, white | 0.767 | 0.004 | 0.004 | 0.038 | 29.2% | 1.3% | 1.3% | 3.7% | 10.3% | 1.2% | 1.2% | 1.2% |
| male, other | 0.836 | -0.002 | 0.000 | 0.017 | 27.9% | -5.0% | 0.0% | 1.3% | 15.4% | -1.6% | 0.0% | 0.0% |
| **Total** | 0.800 | 0.006 | - | - | 28.3% | 0.3% | - | - | 4.7% | 0.2% | - | - |

**Table 3:** Fair use evaluation of a personalized logistic regression model with a one-hot encoding of group attributes for `kidney`. As shown, personalization can improve overall performance while reduces performance for specific groups (red). This result holds across all performance metrics. In such cases, we can resolve fair use violations and improve the gains from personalization by assigning personalized predictions to each group with multiple models. Here, we show the gains when we assign each group the most accurate predictions from either the personalized model $h_g$ or a generic classifier $h_0$, assign each group the most accurate predictions from the personalized model $h_g$ or a decoupled classifier $h^{dcp}$. We highlight cases this intervention led to a gain in green, and cases where it resolved a violation in yellow.

# F  Mortality Prediction for Acute Kidney Injury

In this section, we evaluate the gains of personalization in model to predict mortality for patients with acute kidney injury. We use our results to discuss how fair use evaluations as form of auditing [53] can inform the use of race in clinical prediction models, and describe simple interventions to mitigate harm.

## F.1  Setup

We consider a classification task to predict mortality for patients who receive continuous renal replacement therapy while in the ICU. The data consists of records for $n = 2066$ patients from MIMIC III and IV [38]. Here, $y_i = +1$ if patient $i$ dies in the ICU and $\Pr(y_i = +1) = 51.1\%$. Each patient has $k = 2$ group attributes: `sex` $\in \{$`male, female`$\}$ and `race` $\in \{$`white, black, other`$\}$ and $d = 78$ features related to their health, lab tests, length of stay, and potential for organ failure. We train and evaluate personalized models using the same setup as Section 4.1.

## F.2  Results

We show the performance of a personalized logistic regression model with a one-hot encoding in Table 3, and present results for other model classes in Appendix G. Overall, our findings show that personalization yields uneven gains at a group level. As in Section 4.2, we observe fair use violations across performance metrics and model classes. In this case, for example, the gains in error across range from -5.2% to 6.8%, and two groups experience statistically significant fair use violations: (`male, black`) and (`male, other`).

**On the Use of Race**  Clinical prediction models include group attributes when there is a "plausible" causal relationship between group membership and the outcome of interest. These norms have led to development of widely-used clinical prediction models that use race and ethnicity [25, 71]. Recently, Vyas et al. [71] discuss how these models can inflict harm and urge physicians to check if "race correction is based on robust [statistical] evidence."

Our results highlight how fair use evaluation can provide evidence that serves as a barrier to "race correction" in such cases. Here, checking rationality shows that a race-specific model can reduce performance for specific groups – e.g., (`male, black`) and (`male, other`). Checking envy-freeness reveals that certain groups expect better performance by misreporting their group membership – e.g., (`male, other`) would experience 5.6% gain in test error by reporting any other race.

Even in cases where including race can improve performance, we note that race may act as a proxy for broader social determinants of health. Thus, a model that includes race may act as a "smoke screen" in that it attributes differences in health outcomes to an immutable factor, and perpetuates inaction on the root causes of health disparities [57]. Given these drawbacks, the starting point should be evidence of gain rather than harm.

**F.3    Interventions**

We use our results to simple interventions that can resolve fair use violations by assigning predictions from different models at prediction time. These interventions are admittedly simple, but have the benefit of being broadly applicable.

**Assigning a Generic Model**    We assign groups who are subject to a fair use violation the predictions from a generic model $h_0$. This intervention is guaranteed to resolve all fair use violations in a way that strictly improves performance, and may further reduce the use of personal data in prediction. In this case, it resolves all rationality violations (2/3/2 in terms of error/AUC/ECE respectively). We also observe a potential to reduce data use: seeing how both (`male`, `black`) and (`male`, `other`) experience a fair use violation in terms of error, we see that we could avoid soliciting race for all `male` patients and reduce test error by 1% (as the loss in accuracy for (`white`, `male`) are offset by the gain in accuracy for (`male`,`black`) and (`male`, `other`).

**Assigning a Decoupled Model**    We assign groups who experience a fair use violation the predictions from a *decoupled model* $h_g^{\text{dcp}}$ – i.e., a model fit using only data from their group. While this approach may not resolve fair use violations, it can produce surprisingly large gains as decoupling effectively personalizes the entire model development pipeline. Our results in Table 3 show the potential gains of this intervention across performance metrics. Focusing on error, we see that one can: (1) eliminate fair use violations for 2 groups (`male`,`black`) and (`male`,`other`); (2) greatly improve the gains for 1 group, e.g., (`female`,`black`) who experience a gain of **37.3%**; and (3) improve overall gains by 6.2%. We observe similar effects across other model classes and configurations.

# G    Supporting Material for Sections 4 & F

In this Appendix, we provide: (i) additional information on the datasets used in Sections 4 and F; (ii) results showing the gains of personalization when fitting personalized neural nets and random forests.

## G.1    Additional Information on Datasets

| Dataset | $n$ | $d$ | $\mathcal{G}$ | Prediction Task | Reference |
|---|---|---|---|---|---|
| apnea | 1,152 | 26 | Age × Sex = {<30, 30 to 60, 60+} × {Male,Female} | patient has obstructive sleep apnea | Ustun et al. [66] |
| cardio_eicu | 1,341 | 49 | Age × Sex = {Young,Old} × {Male,Female} | patient with cardiogenic shock dies | Pollard et al. [60] |
| cardio_mimic | 5,289 | 49 | Age × Sex = {Young,Old} × {Male,Female} | patient with cardiogenic shock dies | Johnson et al. [38] |
| heart | 181 | 26 | Age × Sex = {Young,Old} × {Male,Female} | patient has heart disease | Detrano et al. [17] |
| kidney | 2066 | 78 | Sex × Race = {Male,Female} × {White, Black, Other} | mortality of patient on CRRT | Johnson et al. [38] |
| mortality | 21,139 | 484 | Age × Sex = {< 30, 30 to 60, 60+} × {Male,Female} | mortality of patient in ICU | Harutyunyan et al. [34] |
| saps | 7,797 | 36 | Age × HIV = {≤ 30, 30+} × {Positive,Negative} | mortality of patient in ICU | Le Gall et al. [48] |

**Table 4:** Overview of classification datasets used to train clinical prediction models in Sections 4 and F. We describe the conditions that lead to $y_i$ under each prediction task. All datasets used are publicly available, have been deidentified, and inspected to ensure that they contain no offensive content. In cases where data access requires consent or approval from the data holders, we have followed the proper procedure to obtain such consent. Datasets based on MIMIC-III [38] (`kidney`, `mortality`) and eICU [60] (`cardio`) are hosted on PhysioNet under the PhysioNet Credentialed Health Data License. The `heart` dataset is hosted on the UCI ML Repository under an Open Data license. The `apnea` and `saps` datasets must be requested from the authors of the papers listed above [48, 66]. We minimally process each dataset to impute the values of missing points (using mean value imputation), and repair class imbalances across intersectional groups (to eliminate "trivial" fair use violations that occur due to class imbalance).

**apnea**    We use the obstructive sleep apnea (OSA) dataset outlined in Ustun et al. [66]. In this dataset, we have a cohort of 1152 patients where 23% have OSA. We use all available features (e.g. BMI, comobordities, age, and sex) and binarize them, resulting in 26 binary features.

**cardio_eicu & cardio_mimic**    Cardiogenic shock is a serious acute condition where the heart cannot provide sufficient blood to the vital organs. Using the eICU Collaborative Research Database V2.0 [60] and MIMIC-III database [38], we create a cohort of patients who have cardiogenic shock during the course of their intensive care unit (ICU) stay. We use an exhaustive set of clinical criteria

based on the patient's labs and vitals (i.e. presence of hypotension and organ hypoperfusion). The goal is to predict whether a patient with cardiogenic shock will die in hospital. As features, we summarize (minimums and maximums) relevant labs and vitals (e.g. systolic BP, heart rate, hemoglobin count) of each patient from the period of time prior to the onset of cardiogenic shock up to 24 hours. This results in a dataset containing 8,815 patients, 13.5% of whom die in hospital.

**heart**    We use the Heart dataset from the UCI Machine Learning Repository, where the goal is to predict the presence of heart disease from clinical features. It consists of 303 patients, 54.5% of which have heart disease. We use all available features, treating *cp*, *thal*, *ca*, *slope* and *restecg* as categorical, and all remaining features as continuous.

**kidney**    Using MIMIC-III and MIMIC-IV [38], we create a cohort of patients who were given Continuous Renal Replacement Therapy (CRRT) at any point during their ICU stay. For patients with multiple ICU stays, we select their first one. We define the target as whether the patient dies during the course of their selected hospital admission. As features, we select the most recent instances of relevant lab measurements (e.g. sodium, potassium, creatinine) prior to the CRRT start time, along with the patient's age, the number of hours they have been in ICU when CRRT was administered, and their Sequential Organ Failure Assessment (SOFA) score at admission. We treat all variables as continuous with the exception of the SOFA score, which we treat as ordinal. This results in a dataset of 1,722 CRRT patients, 51.1% of which die in-hospital. We define protected groups based on the patient's sex and self-reported race and ethnicity.

**mortality**    We follow the cohort creation steps outlined by Harutyunyan et al. [34] for their in-hospital mortality prediction task. We select the first ICU stay longer than 48 hours of patients in MIMIC-III [38], and aim to predict whether they will die in-hospital during their corresponding hospital admission. As features, we bin the time-series lab and vital measurements provided by Harutyunyan et al. [34] into four 12-hour time-bins, and compute the mean in each time-bin. We additionally include the patient's age and sex as features. This results in a cohort of 21,139 patients, 13.2% of whom die in hospital.

**saps**    The Simplified Acute Physiology Score II (SAPS II) is a risk score that was developed for predicting mortality in the ICU [48]. This study was conducted in 137 medical centers across 12 countries contains 7,797 patients. For each patient we have access to demographics, comorbidities, and vitals which are used to predict the risk of mortality in the ICU. For group attributes we use age and HIV status. The percentage of patients in the dataset who experience mortality is 21.8%.

## G.2   Results for Neural Nets & Random Forests

In this Appendix, we present tables that summarize the gains of personalization for neural networks and random forests. The following tables are analogous to Table 1, except that they also include results for the kidney dataset in Section F.

### G.2.1   Neural Nets

For our neural network models we trained them with two hidden layers of size 5 and 2 and learning rate of $1^{-3}$. Additionally, we applied Platt scaling [59] the outputs of the neural network model to ensure that they were calibrated. We note similar findings described in Sections 4.2 and Section F for neural network models. For example, when looking at test error on cardio_eicu we are able to eliminate all fair use violations by decoupling models. Additionally, across datasets we are able to identify statistically significant fair use violations and gains as noted by the gains and violations rows.

| Dataset | Metrics | Test AUC | | | Test ECE | | | Test Error | | |
|---|---|---|---|---|---|---|---|---|---|---|
| | | 1Hot | All | DCP | 1Hot | All | DCP | 1Hot | All | DCP |
| apnea | Personalized | 0.705 | 0.524 | 0.622 | 6.3% | 2.5% | 5.3% | 36.7% | 50.6% | 41.5% |
| | Gain | -0.012 | -0.193 | -0.095 | -0.7% | 3.2% | 0.4% | -3.3% | -17.2% | -8.1% |
| | Best/Worst Gain | 0.114 / -0.051 | 0.029 / -0.501 | -0.068 / -0.328 | 10.9% / -8.2% | 24.0% / 1.5% | 9.8% / -5.7% | 8.4% / -5.0% | 7.1% / -43.5% | -2.2% / -50.5% |
| | Rat. Gains/Viols | 3/3 | 5/5 | 6/6 | 3/3 | 5/5 | 3/5 | 1/2 | 1/1 | 0/0 |
| | EF Gains/Viols | 1/0 | 0/0 | 2/1 | 2/4 | 5/5 | 5/5 | 6/6 | 6/6 | 6/6 |
| cardio_eicu | Personalized | 0.739 | 0.738 | 0.687 | 4.5% | 5.5% | 5.4% | 31.5% | 31.8% | 36.6% |
| | Gain | 0.001 | -0.001 | -0.051 | 2.3% | 1.4% | 1.5% | 1.6% | 1.3% | -3.5% |
| | Best/Worst Gain | 0.067 / -0.003 | 0.029 / -0.012 | 0.007 / -0.090 | 2.6% / -1.2% | 2.4% / -1.9% | 4.9% / -3.0% | 8.4% / -0.5% | 5.5% / -1.3% | 0.1% / -10.2% |
| | Rat. Gains/Viols | 0/0 | 1/1 | 3/3 | 3/3 | 3/3 | 2/2 | 2/3 | 2/3 | 0/0 |
| | EF Gains/Viols | 0/0 | 0/0 | 2/2 | 1/1 | 2/2 | 2/2 | 4/4 | 4/4 | 6/6 |
| cardio_mimic | Personalized | 0.849 | 0.849 | 0.836 | 3.1% | 4.7% | 3.3% | 23.7% | 24.0% | 23.9% |
| | Gain | 0.004 | 0.004 | -0.009 | 1.1% | -0.4% | 1.0% | 0.6% | 0.2% | 0.4% |
| | Best/Worst Gain | 0.018 / -0.005 | 0.012 / -0.000 | 0.004 / -0.014 | 2.1% / -0.4% | 1.4% / -2.3% | 2.5% / -0.3% | 2.0% / -1.1% | 2.3% / -2.4% | 1.3% / -1.4% |
| | Rat. Gains/Viols | 1/1 | 0/0 | 2/2 | 2/2 | 2/2 | 2/2 | 3/3 | 2/2 | 2/2 |
| | EF Gains/Viols | 0/0 | 1/1 | 4/4 | 4/4 | 3/3 | 3/3 | 4/4 | 4/4 | 4/4 |
| heart | Personalized | 0.457 | 0.736 | 0.554 | 22.7% | 17.2% | 18.1% | 52.6% | 27.6% | 38.2% |
| | Gain | -0.090 | 0.189 | 0.007 | -9.1% | -3.6% | -4.5% | -1.3% | 23.7% | 13.2% |
| | Best/Worst Gain | 0.061 / -0.392 | 0.317 / 0.023 | 0.257 / -0.023 | 4.8% / -29.8% | 9.8% / -9.7% | 6.2% / -14.8% | 1.6% / -9.2% | 38.0% / 4.6% | 28.1% / 7.1% |
| | Rat. Gains/Viols | 2/2 | 0/0 | 2/1 | 1/1 | 1/1 | 2/2 | 0/2 | 4/4 | 3/4 |
| | EF Gains/Viols | 2/1 | 1/0 | 3/1 | 1/1 | 4/4 | 0/0 | 4/4 | 4/4 | 4/4 |
| kidney | Personalized | 0.774 | 0.774 | 0.762 | 6.0% | 6.2% | 7.3% | 29.2% | 31.0% | 30.9% |
| | Gain | 0.003 | 0.004 | -0.009 | -0.1% | -0.4% | -1.4% | -2.1% | -3.8% | -3.7% |
| | Best/Worst Gain | 0.039 / -0.057 | 0.026 / -0.095 | 0.033 / -0.152 | 2.8% / -1.6% | 3.7% / -2.5% | 0.8% / -5.4% | -0.6% / -7.5% | 4.7% / -5.4% | -1.5% / -18.9% |
| | Rat. Gains/Viols | 2/2 | 3/3 | 4/4 | 2/2 | 2/2 | 1/1 | 0/0 | 1/1 | 0/0 |
| | EF Gains/Viols | 1/0 | 1/0 | 4/4 | 3/5 | 3/3 | 2/2 | 6/6 | 6/6 | 6/6 |
| mortality | Personalized | 0.870 | 0.869 | 0.895 | 2.8% | 4.3% | 3.0% | 20.9% | 21.5% | 17.7% |
| | Gain | -0.004 | -0.004 | 0.022 | 0.6% | -0.9% | 0.5% | -0.4% | -1.0% | 2.8% |
| | Best/Worst Gain | 0.025 / -0.019 | -0.001 / -0.015 | 0.039 / 0.005 | 2.7% / -1.7% | 0.5% / -1.2% | 8.0% / 0.1% | 5.1% / -2.2% | -0.3% / -2.3% | 12.6% / 0.1% |
| | Rat. Gains/Viols | 3/3 | 5/5 | 0/0 | 3/3 | 2/2 | 5/5 | 3/3 | 0/0 | 6/6 |
| | EF Gains/Viols | 6/2 | 4/4 | 6/6 | 1/5 | 4/4 | 0/0 | 6/6 | 6/6 | 6/6 |
| saps | Personalized | 0.157 | 0.872 | 0.758 | 37.8% | 7.8% | 31.5% | 63.4% | 21.7% | 48.9% |
| | Gain | -0.037 | 0.678 | 0.565 | 7.5% | 37.5% | 13.9% | -1.8% | 39.9% | 12.7% |
| | Best/Worst Gain | 0.101 / -0.041 | 0.745 / 0.657 | 0.743 / -0.273 | 27.1% / 1.5% | 43.2% / -3.5% | 49.9% / 6.4% | 0.0% / -5.8% | 53.9% / 1.4% | 22.0% / 0.0% |
| | Rat. Gains/Viols | 3/2 | 0/0 | 1/1 | 4/4 | 3/3 | 4/4 | 0/1 | 3/4 | 4/4 |
| | EF Gains/Viols | 3/1 | 3/1 | 0/0 | 3/3 | 3/3 | 3/3 | 4/4 | 4/4 | 4/4 |

**Table 5:** Performance of personalized neural network models on all datasets. We show the gains of personalization in terms of test AUC, ECE, and error. We report: model performance at the population level, the overall gain of personalization, the range of gains over $m$ intersectional groups, and the number of rationality and envy-freeness gains/violations (evaluated using a bootstrap hypothesis test at a 10% significance level).

### G.2.2 Random Forests

For our random forest models, we trained each with the following hyperparameters: 100 estimators, max depth of 20, minimum samples per split is 5, and minimum number of samples in each leaf is 2. For random forests, we expect that these models will perform well when optimizing error but will not necessarily have high AUC or be well calibrated (i.e. low ECE). We note this in the Table below. For example, using an intersectional encoding with random forests in effecitve in minimizing fair use violations on error across multiple datasets (e.g. apnea, kidney). As noted with both logistic regression and neural networks, we are able to reliably identify statistically significant violations.

| Dataset | Metrics | Test AUC | | Test ECE | | Test Error | |
| --- | --- | --- | --- | --- | --- | --- | --- |
| | | 1Hot | All | 1Hot | All | 1Hot | All |
| apnea | Personalized | 0.757 | 0.759 | 7.7% | 7.1% | 30.5% | 31.2% |
| | Gain | 0.002 | 0.004 | -0.4% | 0.6% | 0.8% | -0.5% |
| | Best/Worst Gain | 0.064 / -0.001 | 0.020 / -0.009 | 4.4% / -2.7% | 6.8% / -0.3% | 9.8% / -2.8% | 4.3% / -1.5% |
| | Rat. Gains/Viols | 1/1 | 2/2 | 2/2 | 3/3 | 5/5 | 2/4 |
| | EF Gains/Viols | 4/0 | 4/2 | 4/5 | 3/3 | 6/6 | 6/6 |
| cardio_eicu | Personalized | 0.764 | 0.772 | 7.8% | 8.7% | 30.8% | 30.2% |
| | Gain | 0.001 | -0.000 | -1.3% | -0.7% | 1.0% | 0.4% |
| | Best/Worst Gain | 0.009 / -0.022 | 0.014 / -0.028 | 3.7% / -0.8% | 1.0% / -3.5% | 4.9% / -1.6% | 1.4% / -1.8% |
| | Rat. Gains/Viols | 1/1 | 2/2 | 2/2 | 1/1 | 3/3 | 1/3 |
| | EF Gains/Viols | 1/1 | 3/3 | 3/3 | 3/3 | 4/4 | 4/4 |
| cardio_mimic | Personalized | 0.847 | 0.847 | 9.2% | 9.7% | 24.0% | 24.3% |
| | Gain | -0.003 | 0.001 | 0.4% | -0.4% | -0.2% | -0.3% |
| | Best/Worst Gain | -0.001 / -0.004 | 0.002 / -0.001 | 1.2% / 0.2% | 0.3% / -0.9% | 0.8% / -1.1% | 0.3% / -1.4% |
| | Rat. Gains/Viols | 4/4 | 2/2 | 4/4 | 1/1 | 1/1 | 1/1 |
| | EF Gains/Viols | 2/2 | 2/1 | 1/1 | 3/3 | 4/4 | 4/4 |
| heart | Personalized | 0.897 | 0.896 | 12.0% | 14.4% | 17.1% | 22.4% |
| | Gain | -0.006 | -0.000 | 4.6% | -2.7% | -2.6% | -2.6% |
| | Best/Worst Gain | 0.006 / -0.025 | 0.000 / -0.026 | 5.5% / -1.8% | 9.3% / -5.1% | 9.6% / -10.8% | 5.6% / -10.7% |
| | Rat. Gains/Viols | 3/1 | 4/1 | 2/2 | 1/1 | 1/2 | 0/2 |
| | EF Gains/Viols | 4/0 | 2/0 | 2/2 | 2/2 | 4/4 | 4/4 |
| kidney | Personalized | 0.775 | 0.778 | 8.8% | 9.0% | 29.2% | 29.1% |
| | Gain | 0.001 | 0.003 | -0.7% | -1.1% | 1.2% | 1.0% |
| | Best/Worst Gain | 0.010 / -0.017 | 0.009 / -0.019 | 2.0% / -1.9% | 1.2% / -3.8% | 5.3% / -0.9% | 2.0% / -3.1% |
| | Rat. Gains/Viols | 3/2 | 3/3 | 1/1 | 1/1 | 4/5 | 2/5 |
| | EF Gains/Viols | 3/0 | 3/2 | 4/6 | 3/3 | 6/6 | 6/6 |
| mortality | Personalized | 0.806 | 0.806 | 10.9% | 10.8% | 27.2% | 27.1% |
| | Gain | 0.002 | -0.001 | -0.6% | -0.0% | 0.3% | -0.1% |
| | Best/Worst Gain | 0.005 / 0.001 | 0.012 / -0.004 | 1.4% / -1.2% | 1.5% / -3.1% | 0.9% / -0.5% | 0.7% / -1.8% |
| | Rat. Gains/Viols | 0/0 | 2/2 | 1/1 | 2/2 | 3/3 | 3/4 |
| | EF Gains/Viols | 6/0 | 3/1 | 3/5 | 2/6 | 6/6 | 6/6 |
| saps | Personalized | 0.879 | 0.878 | 4.6% | 4.9% | 20.1% | 20.0% |
| | Gain | -0.002 | -0.002 | 0.0% | 0.1% | -0.4% | -0.4% |
| | Best/Worst Gain | 0.000 / -0.050 | 0.050 / -0.002 | 11.2% / -1.6% | 0.2% / -3.5% | 0.0% / -10.0% | 0.2% / -5.4% |
| | Rat. Gains/Viols | 3/2 | 2/1 | 2/2 | 2/2 | 0/1 | 0/2 |
| | EF Gains/Viols | 3/1 | 4/2 | 4/4 | 4/4 | 4/4 | 4/4 |

**Table 6:** Performance of personalized random forest models on all datasets. We show the gains of personalization in terms of test AUC, ECE, and error. We report: model performance at the population level, the overall gain of personalization, the range of gains over $m$ intersectional groups, and the number of rationality and envy-freeness gains/violations (evaluated using a bootstrap hypothesis test at a 10% significance level).

