# OpenReview forum: "When Personalization Harms: Reconsidering the Use of Group Attributes of Prediction"
_NeurIPS.cc/2022/Workshop/TSRML — TSRML2022_

### Official Review · Reviewer_i3b8 · 2022-10-16
**Good paper with insufficient experiments.**

**Overall Recommendation:** Learning to accept. Details are refer…
**Overall Rating:** 6

**Summary:**

Summary.

This paper is dedicated to discussing the advantages and disadvantages of using group attributes to assign personalized predictions. The authors point out that leveraging group attributes can lead to unnecessary inaccurate predictions for specific groups. They propose several formal conditions and characterize how ML models can exhibit fair use.



**Strengths:**

Pros.

1. The problem setting is very interesting, which detailly dissects the pros and cons of group attributes under different conditions.

2. The visualizations are clear.

3. Sufficient related works are included.

**Weaknesses:**

Cons.

1. Missing the conclusion section.

2. Fonts in Figure 1 and Table 1 are too small to be recognized.

3. Only tiny neural networks and random forests are considered.

**Review Confidence:**

2: The reviewer is willing to defend the evaluation, but it is quite likely that the reviewer did not understand central parts of the paper

---

### Official Review · Reviewer_764a · 2022-10-20
**Thorough Investigation of Group Attributes in Medical Models**

**Overall Rating:** 8

**Summary:**

This paper:
1) Explores how using protected attributes can lead to a decrease in performance for groups.
2) Sets a clear definition for the minimum quality of service that should be had for models that are trained using group attributes.
3) Provides examples and case studies where this phenomenon happens in practice.
4) Explores modeling design decisions that impact the performance discrepancy, and how they can be used to mitigate the results.

**Strengths:**

This paper is very clear and complete. It is very easy to follow, and does a complete analysis of its problem statement. It provides a clear example of where systems can be unfairly biased against different groups. It shows this principle in both toy examples and real-world situations. It also provides reasoning for why this bias may occur, and multiple methods to help mitigations the bias. Societally, it gives practitioners something to consider when building models, and also provides them with tools to use to help address the problem.

**Weaknesses:**

The paper explores the use of different model types, and methods of encoding features of the subject and their protected group attributes. It addresses the performance of the model as a whole and on the different protected groups using the various methods. In many of the communities that these types of models are (e.g. finance) there is a strong emphasis on model interpretability - and making it transparent how the score was calculated. It would be nice if the authors were to address how some of the alternative techniques that they propose effect the ease of interpretability.

Additionally, as there was no clear consensus as to which method worked the best to prevent lower performance for protected groups, it would be nice if the authors mentioned how to avoid overfitting to the design choice.

**Overall Recommendation:**

This paper is a clear accept. It fits well into the Trustworthy and Responsible Machine Learning, as it shows a real-world, yet not-yet explored, example of unintentional harm that machine learning models can have against certain groups. It provides a clear situation where this bias occurs, a clear method of determining whether or not a model is safe to use, and several examples of how to mitigate the harm.

**Review Confidence:**

4: The reviewer is confident but not absolutely certain that the evaluation is correct

---

### Decision · Program_Chairs · 2022-10-23

Accept